# The Curious Case of the “Neurotoxic Skink”: Scientific Literature Points to the Absence of Venom in Scincidae

**DOI:** 10.3390/toxins13020114

**Published:** 2021-02-03

**Authors:** Kartik Sunagar, Siju V Abraham

**Affiliations:** 1Evolutionary Venomics Lab, Centre for Ecological Sciences, Indian Institute of Science, Bangalore 560012, Karnataka, India; 2Department of Emergency Medicine, Jubilee Mission Medical College and Research Institute, Thrissur 680005, Kerala, India; sijuvabraham@jmmc.ac.in

**Keywords:** venom evolution, the origin of reptilian venom, neurotoxic venoms in skinks

## Abstract

In contrast to the clearly documented evolution of venom in many animal lineages, the origin of reptilian venom is highly debated. Historically, venom has been theorised to have evolved independently in snakes and lizards. However, some of the recent works have argued for the common origin of venom in “Toxicofera” reptiles, which include the order Serpentes (all snakes), and Anguimorpha and Iguania lizards. Nevertheless, in both these contrasting hypotheses, the lizards of the family Scincidae are considered to be harmless and devoid of toxic venoms. Interestingly, an unusual clinical case claiming neurotoxic envenoming by a scincid lizard was recently reported in Southern India. Considering its potentially significant medicolegal, conservation and evolutionary implications, we have summarised the scientific evidence that questions the validity of this clinical report. We argue that the symptoms documented in the patient are likely to have resulted from krait envenomation, which is far too frequent in these regions.

## 1. Introduction

Traditionally, an anthropocentric interpretation of venoms has led to the consideration of only animals capable of inflicting clinically severe symptoms in humans as “venomous”, whereas the ecological roles of these toxic secretions have been largely neglected. Over the past few decades, however, investigations into the impact of ecology and environment in shaping animal venoms have gained momentum [1]. Several studies have now demonstrated the roles of diet, ontogeny, predator pressure, intraspecific competition, geographic distribution, and climatic conditions in shaping animal venoms [2,3,4,5,6]. For instance, a relatively reduced or complete loss of toxicity towards mammals has been shown in certain snakes that chiefly feed on non-mammalian prey systems [6,7,8], making them ineffective or relatively less effective against humans. The modern definition, therefore, defines venom as a specialised form of poison that is actively injected to facilitate the quotidian functions of the venomous animal, including predation, self-defence and intraspecific competition.

Considering their clinical relevance to humans, snake venoms have been rigorously investigated to date. In complete contrast, the evolutionary diversification of venom and venom delivery systems (VDS) in lizards remains relatively underexplored. While certain species of lizards, such as the Gila monster (*Heloderma suspectum*) and the Mexican beaded lizard (*H. horridum*), are believed to be capable of inflicting clinically severe envenoming to humans, and a few others have been suggested to employ venom for predation [9], the presence of venom in most other lizard lineages is largely undocumented. Nonetheless, skinks of the family Scincidae are feared in the rural regions of the Indian subcontinent and are often unnecessarily killed on sight. In support of this fallacious myth, a peculiar clinical case of neurotoxic envenoming by the common dotted garden skink (*Riopa punctata*; incorrectly referred to as “the red-tailed skink [*Lygosoma punctatum*]” by the authors) was reported from southern India [10]. Considering the potential implications of this study on our understanding of reptilian venom evolution, as well as its negative impact on conservation, we summarise the scientific evidence that points to the absence of venom in this enigmatic lineage of lizards and refutes the validity of this clinical report.

## 2. The Evolutionary Origin of “Toxicofera” Venom Systems

Given their significant role in underpinning the evolutionary success of many animal lineages, venoms are a fascinating system for research in evolutionary biology. The origin and diversification of venom, in particular, has been widely investigated. While over a hundred independent origins of venom across the breadth of the animal kingdom have been traced, the precise origin of venom in reptiles is still debated. Historically, venom is theorised to have independently evolved twice in reptiles: once in Caenophidia or advanced snakes (Elapidae, Viperidae, and non-front-fanged “Colubridae”) and on another time in *Heloderma* lizards (Figure 1) [11,12,13]. In contrast to this, a single early origin of venom in the common ancestor of the order Serpentes, and Anguimorpha and Iguania lizards, which constitutes the hypothetical “venomous” clade, Toxicofera, has also been postulated recently [14,15]. Nonetheless, given the lack of evidence for the presence of venom in Scincidae, this family is not considered venomous by either of these hypotheses.

## 3. Evidence Supports the Lack of Venom in Scincidae

While skinks are feared in the rural regions of India, there are multiple lines of evidence that support the lack of venom and efficient VDS in this lineage [15]. By employing a multi-pronged strategy, involving venom proteomics, venom gland transcriptomics, evolutionary analyses and histological and anatomical assessments, it has been demonstrated that lizards outside the “venom clade” have poorly developed VDS [15]. Venomous snakes and lizards were largely shown to possess an increased number of protein secreting “serous” cells in their oral glands, whereas non-toxicoferan lizard salivary glands, including those of Scincidae, were found to be enriched in mucous secreting cells. This strongly suggests that skinks are incapable of producing large amounts of proteins and delivering them efficiently into their bite victim. Moreover, the proteomic characterisation of salivary secretions and transcriptomic profiling of skink salivary glands did not recover any of the known toxin homologues that enrich the venoms of Toxicofera reptiles [15]. A recent study examining the abilities of lizard salivary secretions to induce coagulopathies on the human blood convincingly demonstrated that, unlike *Heloderma* and varanid lizards, skink saliva does not show any effect [16]. Furthermore, given the lack of strong evidence confirming the presence of venom, some reports have even argued for the absence of venom in certain lizards within the “Toxicofera” clade [17]. These findings collectively point to the lack of a venom arsenal in Scincidae.

## 4. Potential Causes of the Observed Neurotoxic Envenoming

In the clinical report under discussion [10], the patient may have witnessed the skink bite, but the culprit animal responsible for the neuromuscular paralysis is still questionable. Particularly, whether the animal that was produced by the patient was the only one that bit them cannot be stated with certainty. Snakebite, a neglected tropical disease, is prevalent in all parts of India, including Tamil Nadu [18]. While a large number of envenoming in the country is attributed to the “big four” Indian snakes, including the spectacled cobra (*Naja naja*), Russell’s viper (*Daboia russelii*), common krait (*Bungarus caeruleus*) and saw-scaled viper (*Echis carinatus*) [6], the culprit snakes are never identified in 40 to 50% of cases [19,20]. Krait bites, in particular, are notorious for being painless and elusive [21]. The literature shows that approximately 10% of the identified venomous snakebites in the region where this clinical case was reported (Tirunelveli) were from kraits [20]. Certain species of kraits, including *B. caeruleus*, are not only known to feed on mammals, but also on reptiles, amphibians and birds [22,23], which has been shown to significantly impact their venom compositions and potencies [6]. Considering the fact that a skink, a putative prey species of *B. caeruleus*, was found near the victim, and isolated neurotoxic symptoms were observed in the patient, the possibility of krait envenoming cannot be ruled out. Moreover, these neurotoxic symptoms did not subside, despite the administration of neostigmine, which is strongly suggestive of presynaptic toxicity, possibly resulting from krait envenoming. The other possible differentials of acute flaccid neuromuscular weakness, with or without presynaptic involvement, may include botulinum poisoning, rare inherited channelopathies (e.g., hypokalemic paralysis) and autoimmune disorders (e.g., Lambert-Eaton Myasthenic Syndrome, Guillain-Barré Syndrome (GBS) and its variants) [24,25]. Interestingly, neurotoxic envenoming by snakes and GBS have been documented as the two most common causative agents of acute flaccid neuromuscular paralysis in some parts of the country [26]. Therefore, rather than presuming the existence of a potent neurotoxic venom in skinks, when no such evidence exists to date, it would be prudent to investigate this case report from an epidemiological perspective, as well as considering all other differential causes of the signs and symptoms observed.

## 5. A Typical Case of “Early Morning Neuroparalytic Syndrome”

Syndromic classification of snakebites, although not without its limitations [27], is a useful approach to consider, as it facilitates the reliable diagnosis of snakebites in most cases without relying solely on live or killed snake specimens that are brought along by the bite victim [28]. This is also advantageous as the abilities of snakebite victims, bystanders, and healthcare providers to spot and identify the culprit snake species is context-specific and often unreliable [29]. Syndrome-species correlation studies have revealed the accuracy of relying on the major syndromes in identifying envenomings by the ‘big four’ medically important snakes in India [19,27]. It should, however, be noted that identifying the culprit snake is not essential, as a single polyvalent antivenom is utilised for the treatment for snake envenoming throughout India. The patient’s clinical history, signs and symptoms (i.e., sleeping on the floor, absence of hemorrhagic manifestations, absence of swelling at the bite site, bilateral ptosis, shallow respiration rate and, ultimately, respiratory failure, with little improvement with neostigmine) reported in this clinical study [10] largely point to krait envenoming [27,28,30]. Moreover, snakebite induced neuroparalysis, which is often termed as the “early morning neuroparalysis” or “slum (or jhuggi) dwellers syndrome”, has been largely documented in people waking up from sleep on the floor [31,32,33]. Thus, it is highly probable that the observed neuroparalysis in the patient was inflicted by a krait species, and not by a harmless skink.

Lizards have been rarely documented to cause clinically severe envenoming in humans. Similarly to the case of the “neurotoxic skink” [10], the common Indian monitor lizard (*Varanus bengalensis*) bite was allegedly reported to cause coagulopathy, haemolysis, rhabdomyolysis, sepsis, and fatal acute kidney injury due to a temporal association of the bite with the presence of a varanid lizard [34]. However, considering that these effects could also result from a snakebite, especially by the medically important vipers prevalent in the region, this case report was later vehemently criticised [35]. As extraordinary claims warrant extraordinary evidence, it is important to rigorously investigate the patient’s history and gather scientifically robust data to support such assertions. Venom proteomics, venom gland transcriptomics, evolutionary analyses and histological and anatomical assessments of toxicoferan and non-toxicoferan reptiles have previously demonstrated that the lizards of the family Scincidae are unlikely to be venomous [15]. Homologues of toxic venom proteins have never been identified from skink saliva, and the composition of their oral gland cells suggests that they are unlikely to secrete a toxic cocktail and deliver them in concentrations required to cause life-threatening envenoming in humans. Thus, it is not surprising that there are no conclusive cases of neurotoxic envenoming by skinks. Considering the existing body of knowledge in venom biology, which points to the absence of venom in Scincidae, in conjunction with the clinical signs and symptoms observed in the patient, and the epidemiology of envenoming in the region, conclusions presented in the clinical case report [10], attributing the presence of a hitherto unknown “neurotoxic venom” to *R. punctata*, are questionable.

## Figures and Tables

**Figure 1 toxins-13-00114-f001:**
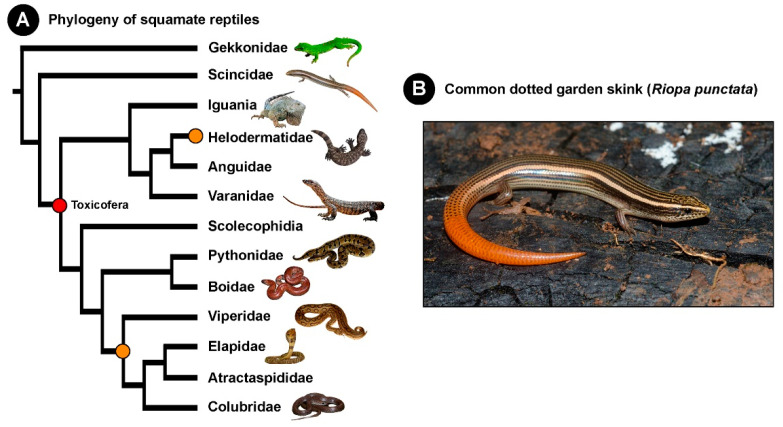
This figure depicts (**A**) the phylogeny of squamate reptiles with the single (red sphere) and dual (orange sphere) origins of venom indicated; and (**B**) the photograph of the common dotted garden skink (*Riopa punctata*).

## Data Availability

Data sharing not applicable.

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
