# Peer review of "The Curious Case of the “Neurotoxic Skink”: Scientific Literature Points to the Absence of Venom in Scincidae"

_toxins, 2021, doi:10.3390/toxins13020114_

Round 1

Reviewer 1 Report

In this comunication the authors discuss the evolution of venom production and secretion among Reptilia based on a case discription of an Indian lizard. The communication is important due to the philogenetic and proteomic literature on Reptilia, supporting the hypothesis that the Scincidae lyzards are not expected to be venomous and that the published data should be experimentally reviewed.

Author Response

We thank the reviewer for their supportive comments.

Reviewer 2 Report

The authors present a communication questioning a report on a neurotoxic skink bite published in the Journal of The Association of Physicians of India. The authors provide alternatives for the symptoms and ultimate death of the patient with the primary one being that the bite was actually from a Krait. 

I think the need for this communication is there because it is best to immediately question reports that have no data supporting them. Based on the evidence the authors present, it is implausible to impossible that the skink has a neurotoxic bite which is in line with what is know about Scincidae. However, the authors used much of the communication to add unnecessary information about the broader context of venom and venom literature without fully addressing the paper that they are rebutting. 

Additionally, the title makes it seem like the skink is actually neurotoxic. Overall, the report needs to be streamlined and the majority of the information in Section 1 and Section 2 can be removed in favor of summarizing the paper that is being refuted and at least providing the symptoms that the woman presented. 

Lastly, please ensure the abstract focuses on the impetuous for the communication and the likely causes of the woman's reaction being from a krait.

Author Response

The authors present a communication questioning a report on a neurotoxic skink bite published in the Journal of The Association of Physicians of India. The authors provide alternatives for the symptoms and ultimate death of the patient with the primary one being that the bite was actually from a Krait.

I think the need for this communication is there because it is best to immediately question reports that have no data supporting them. Based on the evidence the authors present, it is implausible to impossible that the skink has a neurotoxic bite which is in line with what is know about Scincidae.

We thank the reviewer for their kind and supportive comments.

However, the authors used much of the communication to add unnecessary information about the broader context of venom and venom literature without fully addressing the paper that they are rebutting. Overall, the report needs to be streamlined and the majority of the information in Section 1 and Section 2 can be removed in favor of summarizing the paper that is being refuted and at least providing the symptoms that the woman presented.

We respectfully disagree with the reviewer on the first part of this comment. We believe that the evolution of venom is the major argument that points to a lack of venom in skinks. Hence, it is crucial to provide this context and background to clinician colleagues, as many may not be aware of the origin and evolution of reptilian venom. Moreover, we have discussed every relevant point that was made in the original clinical report, including all the symptoms that were reported. Please see below.

“The patient’s clinical history, signs and symptoms (i.e., sleeping on the floor, absence of haemorrhagic manifestations, absence of swelling at the bite site, bilateral ptosis, shallow respiration rate and, ultimately, respiratory failure, with little improvement with neostigmine) reported in this clinical study [10], largely point to krait envenoming [27,28,30].”

Additionally, the title makes it seem like the skink is actually neurotoxic.

We thank the reviewer for this comment and, following their suggestion, we have now modified the title to: The curious case of the “neurotoxic skink”: Scientific literature points to the absence of venom in Scincidae

Lastly, please ensure the abstract focuses on the impetuous for the communication and the likely causes of the woman's reaction being from a krait.

We thank the reviewer for this suggestion. We have now added the following line to clarify the same.

“We argue that the symptoms documented in the patient are likely to have resulted from krait envenomation, which is far too frequent in these regions.”

Reviewer 3 Report

I congratulate the authors of this concise commentary on their judicious and balanced consideration of this highly controversial case report. Obviously, using a Popperian framing, they might also invite formal studies of skink saliva LD50 values to attempt a falsification of their assertion of lack of skink 'venom' toxicity. As the saying goes, extraordinary claims warrant extraordinary evidence.

It would be helpful, to the non-herpetologists amongst the readership, to provide a summary table of skink and other non-toxicoferan reptile data so discussed. That is to say, the authors cite a single paper (ref #15) as the major source of evidence against non-toxicoferan reptile 'venom' - how many skink species were included ? Geographical origin ? Number of individual specimens examined ? Greater documentation would assist in non-expert reader in assessing the comprehensiveness of this critical citation.

Author Response

I congratulate the authors of this concise commentary on their judicious and balanced consideration of this highly controversial case report. Obviously, using a Popperian framing, they might also invite formal studies of skink saliva LD50 values to attempt a falsification of their assertion of lack of skink 'venom' toxicity. As the saying goes, extraordinary claims warrant extraordinary evidence.

We thank the reviewer for their very kind comments. We completely agree with their comments and this was, indeed, our intention.

It would be helpful, to the non-herpetologists amongst the readership, to provide a summary table of skink and other non-toxicoferan reptile data so discussed. That is to say, the authors cite a single paper (ref #15) as the major source of evidence against non-toxicoferan reptile 'venom' - how many skink species were included ? Geographical origin ? Number of individual specimens examined ? Greater documentation would assist in non-expert reader in assessing the comprehensiveness of this critical citation.

While we fully agree with this comment, venom gland transcriptomics studies rarely assess more than a single specimen as this requires animal euthanisation. As most reptiles around the world are protected by the law, it is extremely difficult to euthanise multiple individuals for a study. The only skink that was euthanised for examination in the cited paper was captive bred and its geographical origin was unknown. We had initially cited a single paper as it was the only paper where skink saliva and salivary glands have been examined via transcriptomics, proteomics and histological experiments. We have now cited another recent paper, where the skink saliva was examined for its ability to induce coagulopathies in human bite victims. This too supports our argument that the skink may not have been responsible for the symptoms documented in the patient. To date, there are only two publications that have assessed the salivary secretions of skinks. In contrast, there have been many studies where the venom and venom glands of other reptiles, including the closest relatives of Scincidae, have been assessed. Hence, we feel that it may not be relevant to include such a table.

Round 2

Reviewer 2 Report

I thank the authors for addressing the comments of the three reviewers.